

# A smartphone microscopic method for rapid screening of cloth facemask fabrics during pandemics

Bhanu B. Neupane[1], Ravindra K. Chaudhary[2] and Amita Sharma[3]

[1] Central Department of Chemistry, Tribhuvan University, Kathamndu, Nepal
[2] Amrit Campus, Department of Chemistry, Tribhuvan University, Kathmandu, Nepal
[3] Center for Analytical Sciences, Kathmandu Institute of Applied Sciences, Kathmandu, Nepal

## ABSTRACT

**Background:** In pandemics, because of increased demand and subsequent shortage of commercial facemasks, people need to use cloth facemasks, although such masks are reported to provide reduced protection. These masks can be prepared in local levels from different fabric materials. In developing countries, cloth masks are preferable because of low cost and added advantages of reusability. The filtering performance of a cloth facemask depends on the facial fit and on the material properties of fabrics such as porosity, yarn spacing or packing, and pore size. In resource limited settings, an affordable and easy to implement method that can assess the surface properties of cloth facemask fabrics would be important.
**Methods:** In this work, we developed a smartphone microscopic method for rapid screening of fabric quality. We measured the field of view of the microscope and as a proof of concept, we implemented the method to examine surfaces of sixteen locally available cloth mask fabrics.
**Results:** Out of the 16 masks examined, we found very diverse yarn packing and pore morphology (pore size and shape) in the fabrics. The pore size ranged from ~80 to 720 μm; much larger than respiratory droplet and bio-aerosol. This observation partly explains why such cloth facemasks provide reduced protection to the user during pandemics. The performance of a cloth facemask partly depends on the material properties of fabric such as yarn packing, pore size, porosity. Therefore, the surface properties of fabrics obtained from the smartphone method can be used to get preliminary idea on the facemask quality. We believe that the method can be an affordable and rapid method for selection of better fabrics for cloth facemask during pandemics.

## INTRODUCTION

Facemasks and respirators are important components of personal protective equipment for containing bio-aerosol and droplet mediated transmission of a disease. The filtering efficiency of a filtering device depends on the nature of filter media, size of particle, and environmental conditions (*Barrett & Rousseau, 1998*; *Hutten, 2015*;

Corresponding author
Bhanu B. Neupane,
bbneupane@cdctu.edu.np

*Neupane & Giri, 2020*). The level of protection a filtering device provides also depends on the user compliance and the facial seal (*Bard et al., 2019*). To prevent the human-to-human transmission of a disease, for example COVID-19, a proper filtering device should be worn and other infection and prevention control measures should be followed. Also, special precaution should be followed while wearing and disposing a contaminated mask (*World Health Organization, 2020*, p. 19). In resource limited settings and during outbreaks, because of increased demand and subsequent shortage of commercial masks (*Ha, 2020*), facemasks made from locally available fabric material are also widely used. Cloth facemasks are preferable due to low cost and the added advantages of reusability and lower breathing resistance (*Chughtai, Seale & MacIntyre, 2013*).

Studies on particulate matter filtering performance of two layered cloth facemasks have reported that such masks provide poor filtration efficacy to the user (*Van der Sande, Teunis & Sabel, 2008*; *Rengasamy, Eimer & Shaffer, 2010*; *Shakya et al., 2017*). A study on aerosol filtering efficiency of common fabrics and their combination is reported in a recent study and recommended that fabrics having high yarn packing and low porosity along with multi-layered design is required for better performing cloth facemask (*Konda et al., 2020*). Study on virus filtering efficiency in laboratory settings is also reported in literature. The cloth facemasks, depending on the type of fabrics, is reported to have virus (bacteriophage MS2) filtering efficiency of 50–70% (*Davies et al., 2013*). The virus filtering efficiency of three ply SMS (Spun bonded–Melt blown–Spun bonded) type surgical masks and N95 respirators was reported to be 85–95%(*Balazy et al., 2006*; *Davies et al., 2013*) and 95–97% (*Balazy et al., 2006*, p. 95; *Harnish et al., 2016*; *Rengasamy et al., 2017*; *Zhou et al., 2018*), respectively. A cluster randomized trial study on the effectiveness of cloth masks (*Davies et al., 2013*), medical masks (*Leung et al., 2020*) and respirators (*Radonovich et al., 2016*) in hospital settings was also reported. It was found that the influenza like illness was higher in health care personnel who wore cloth facemask than those who wore surgical facemask (*MacIntyre & Chughtai, 2015*). This study suggested that cloth facemasks cannot be recommended in hospitals settings. Nonetheless, in public places during pandemics, where social distancing is difficult to maintain and commercial masks are not available, it is recommended that a cloth facemask be used (*CDC, 2020*).

The poor performance of cloth masks is due to improper facial fit and inherent properties of fabrics used. If a fabric having high yarn or thread packing and low porosity (or high cover factor), for example a tightly weaved cotton having 600 threads per inch, is used in designing a facemask then user can get better filtration efficacy (*Konda et al., 2020*). The facemasks designed from fabrics having high yarn packing and smaller pores perform better than masks designed from fabrics having larger pores (*Neupane et al., 2019*). This means that selection of proper fabrics is important in designing a better performing cloth facemask. In pandemics and in resource limited settings, standard filtering efficiency measurement setup is difficult to access; therefore, an affordable and easy to implement method that can guide the mask user and or designer for selection of better fabrics for cloth facemask would be important.

In this work, we developed a portable smartphone microscope system that can image the cloth facemask fabrics in bright field mode. We measured the field of view of the

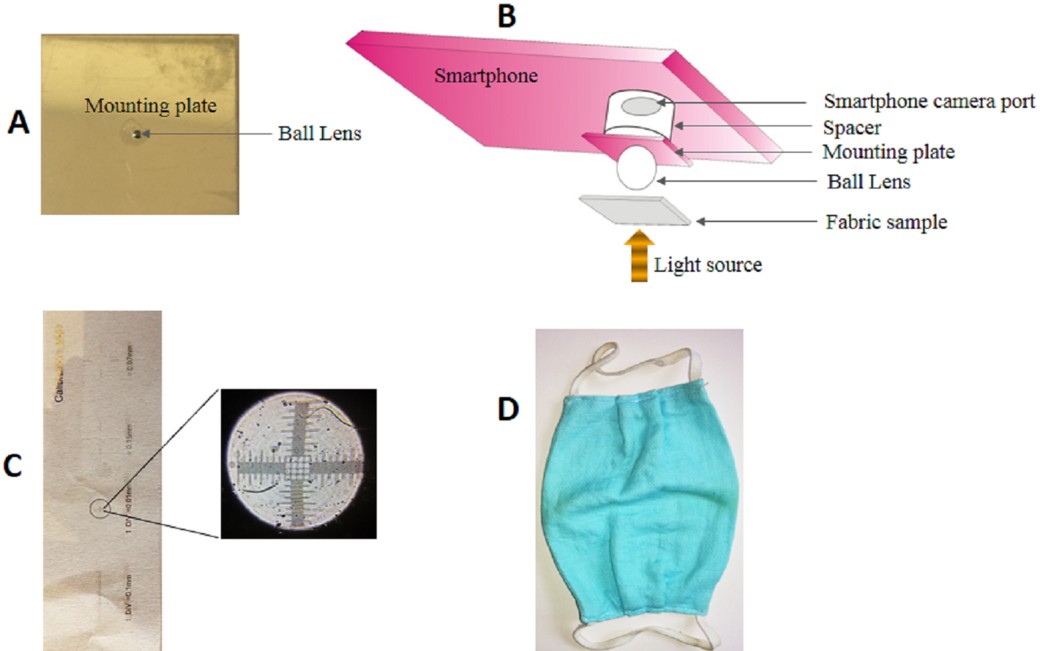

**Figure 1 Components of a smartphone microscope.** (A) A 4 mm spherical ball lens mounted on an aluminum plate. (B) Schematics of the optical setup of the smartphone microscope. (C) Photograph of a calibration slide. The inset is the smartphone microscope image of the specified region within the slide to show the linear grid pattern. (D) A representative image of a cloth facemask used in this study.

microscope and applied it to image the surface of 16 two-layered cloth facemasks. The optical images were analyzed to get information on yarn packing and pore size. A guideline for selection of better fabrics for cloth facemask during pandemics is also provided.

# MATERIALS AND METHODS

## Design of microscope system

The spherical sapphire ball lens of 4 mm diameter was purchased from Edmond Optics, USA (unit price ~$20). An aluminum plate (2 × 2 × 0.1 cm) was obtained from a local machine shop and a hole of ~3.5 mm diameter was made at the center of the plate. The ball lens was fixed on the hole with the help of transparent nail polish (Fig. 1A). The assembly was then mounted on a smartphone camera (3120 × 4160 pixels, screen size 12.2 × 6.7 cm) with the help of double sided transparent tape. The working distance between smartphone camera and ball lens assembly was adjusted by inserting a piece of a paper cardboard (2 × 2 × 0.1 cm) having a hole of around 0.8 cm diameter at the center. The light source needed for illumination was made by putting a white LED in closed cardboard box of 15 × 15 × 15 cm dimension that had a 0.5 cm hole (called illuminating hole) on the top center of the box. The whole assembly is called smartphone microscope. A simple schematics of the microscope is shown in Fig. 1B.

To measure the field of view of the microscope, a calibration slide (Fig. 1C) having minimum inter-grid distance of 10 μm was placed on the top of light box pinhole and

**Table 1 Information on the cloth facemask fabrics.** L1 and L2 indicate the two fabric layers in type II masks.

| Masks | Type of fabric | Masks | Type of fabric |
|-------|----------------|-------|----------------|
| M1 | polyester-spandex | M9 | polsyeter-spandex |
| M2 | cotton | M10 | polyester-spandex |
| M3 | cotton-spandex | M11 | cotton |
| M4 | rayon-spandex | M12 | cotton |
| M5 | cotton-spandex | M13 | cotton (L1) and polyester-spandex (L2) |
| M6 | cotton | M14 | polyester-spandex (L1) and rayon-spandex (L2) |
| M7 | cotton-spandex | M15 | cotton-spandex (L1) and polyester (L2) |
| M8 | cotton-polyester | M16 | cotton (L1) and cotton-spandex (L2) |

imaged onto the smartphone camera. The number of grids in the central focused region in the image were counted to get the field of view in micrometer.

The field of view (FOV) of an imaging system is the maximum area that can be observed in a camera or in an eyepiece (*Stender et al., 2013*). The FOV of a ball lens depends on its diameter, refractive index of the lens material ($n$), and wavelength of light used ($\lambda$). The FOV increases with the increase of ball lens diameter. Theory predicts that the FOV for 4 mm sapphire ball lens ($n = 1.77$) at $\lambda$ of 0.55 μm is around 740 μm (*Cybulski, Clements & Prakash, 2014*).

The spherical ball lens has curved surface. This results in curvature effect that is rays incident on the lens surface are not focused at the same image plane. Because of this effect only the central region of the image can be used for study. If balls lens is attached to the smartphone camera, the FOV is also partly determined by smartphone camera specifications. The effective FOV of our smartphone microscope system was 760 ± 5 μm.

## Imaging of face mask

Cloth face masks were purchased from local markets (unit price $0.2–0.3). All masks had two layers and stretchable ear loops (Fig. 1D). The masks, however, had same or different fabrics in two layers. The masks having same and different fabrics in two layers are hereunder labeled as type I and II masks, respectively. A total of 12 type I (labeled M1–M12) and four type II (labeled M13–M16) masks were selected for the study. For each mask M1–M16, three replicas were considered. The information on the type of fabrics used in mask is provided in Table 1. The masks containing spandex were more stretchable than others.

For imaging, a small piece of mask (50 mm × 10 mm) was cut with scissors without stretching the mask surface. The piece was placed flat on a clean glass slide. The slide was then placed on the top of light box hole and imaged with the smartphone microscope. We did preliminary inspection of all three replicas with the smartphones microscope and found their images similar. So, we saved the image of only one of the replica and did further analysis. Around 5–10 images were collected in the different regions of the mask for each layer by manually scanning the smartphone microscope. The collected images

were transferred to computer and imported to ImageJ software (NIH, Bethesda, MD, USA). The pixel number was converted to micrometer to get the pore size information. To get the information on yarn packing we calculated the total cover factor of fabric ($f$) that is, the % of total area of fabrics covered by yarns, which is defined as:

$$f = \left(\frac{A_{\text{yarn}}}{A_{\text{total}}}\right) \times 100 \tag{1}$$

where, $A_{\text{yarn}}$ is total area covered by yarns in the fabric and $A_{\text{total}}$ is the total area covered by the fabric or total accessible area. The approximate cover factor was obtained from the microscopic images of the fabrics as (*Tàpias, Ralló & Escofet, 2011*):

$$f' = \left(\frac{n_{\text{yarn}}}{n_{\text{total}}}\right) \times 100 \tag{2}$$

where, $n_{\text{yarn}}$ is total number of pixels corresponding to yarns and $n_{\text{total}}$ is the total number of pixels in the fabric image. Manual thresholding was used to classify the pixels as dark pixels (that correspond to yarn) and bright pixels (pores) in the ImageJ software. The opposite term of cover factor is porosity and was calculated as:

$$\text{Porosity} = 100 - f' \tag{3}$$

## RESULTS

### Imaging the surface of type I masks

The images of all the masks were collected in bright field mode, so the bright patches on the image represent the pores and the dark regions the yarns (Fig. 2; Fig. S1). The pores in the fabric surface, except in the M6 and M12 masks, are clearly visible and mostly asymmetric. This creates a challenge to measure dimension of the pore. Nonetheless, we measured the longest dimension of the pores in the fabric to get the approximate upper estimate of the pore size. It is interesting to see the pores of diverse morphology, yarns of different diameter, and also the difference in yarn packing in the fabrics of different masks. The pores in M6 and M12 are not easily visible, but we adjusted the image contrast and measured the size of brightest region in the image to estimate the pore size. Since the pores are not obvious, the pore size estimation obtained from contrast adjustment in M6 and M12 masks may have more error.

To get quantitative information on yarn packing, we also calculated the % of total area covered by yarns (cover factor) for the fabrics in all the masks by using the methodology provided in experimental section (Eq. 2). The pore size and cover factor data for the fabrics in type I masks is provided in Table 2. It is interesting to see the pore size in the range of ~77 (in M6) to ~460 µm (in M10); larger than the typical size of respiratory droplets (5–100 µm). The cover factor ranged from ~66 (in M10) to ~96% (in M12).

### Imaging the surface of type II masks

The images of type II masks were also collected in bright field mode. The type II masks had different fabrics in two layers, so the difference in surface of fabrics in two layers is very

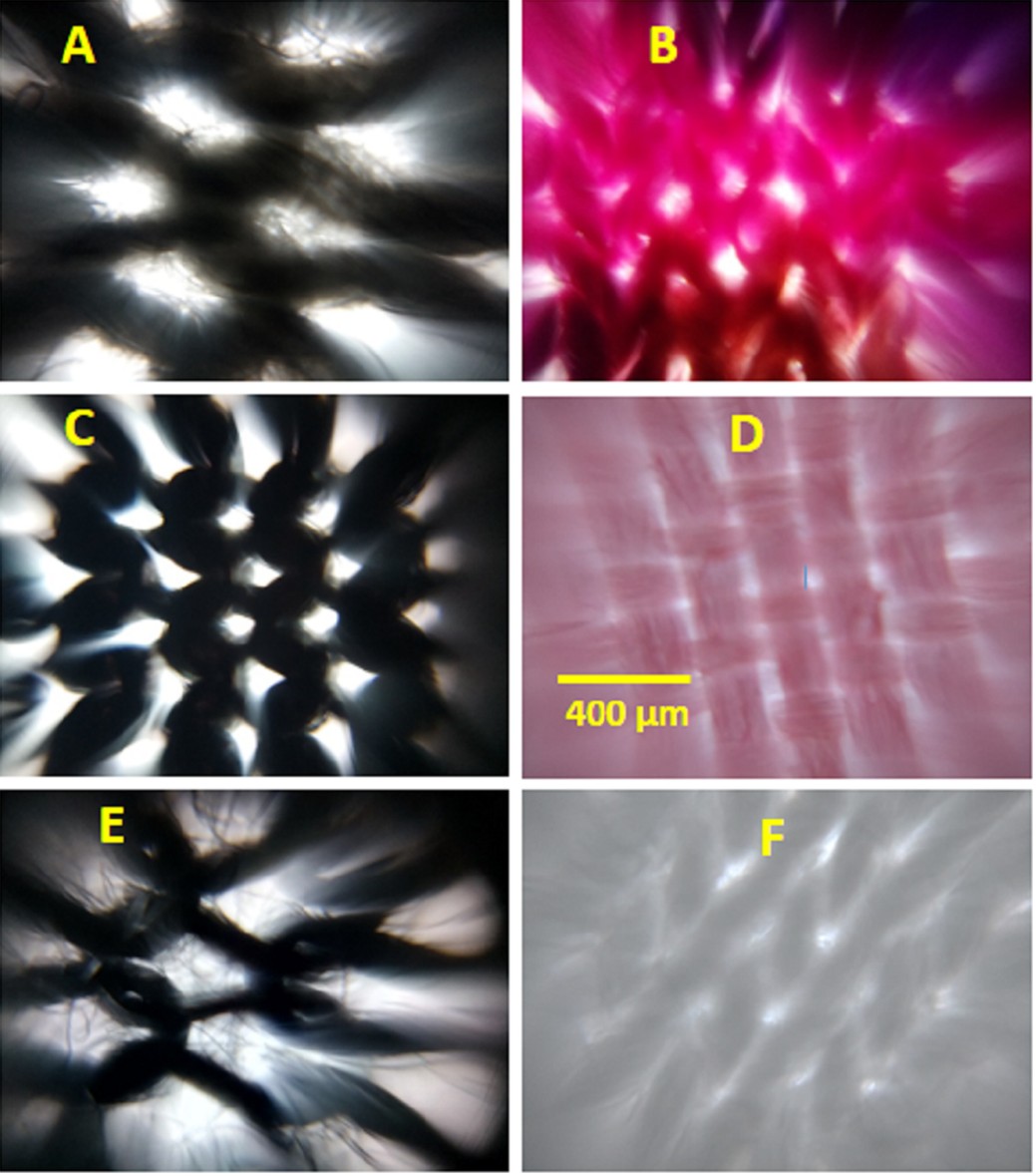

**Figure 2** **Representative images of some of the type I cloth facemask.** (A) M1, (B) M2, (C) M5, (D) M6, (E) M10, and (F) M12. The scale bar shown in D is 400 μm and applies to all images.

obvious in the images shown in Fig. 3 and Fig. S2. The pore size in the layer I of M16 are not easily visible, but we adjusted the image contrast and measured the size of brightest region within the image to get a rough estimate of pore size. Again as in M6 and M12 masks, the pore size estimation obtained from contrast adjustment can have more error. The pore size and cover factor data for both fabrics layers (L1 and L2) in the type II masks is provided in Table 3. It is interesting to find the pore size in layer II of M15 are as big as 720 μm and cover factor as low as 42%. As in type I masks, pores in all fabrics layers of type II masks are larger than the typical size of respiratory droplets.

**Table 2 Pore size and cover factor in fabrics of type I masks.** The mean pore size and cover factor was calculated form ten and five measurements, respectively.

| Masks | Pore size (mean ± σ) μm | Cover factor (mean ± σ) | Masks | Pore size (mean ± σ) μm | Cover factor (mean ± σ) |
|---|---|---|---|---|---|
| M1 | 394 ± 38 | 78 ± 4 | M7 | 204 ± 23 | 82 ± 3 |
| M2 | 167 ± 26 | 87 ± 2 | M8 | 197 ± 27 | 87 ± 1 |
| M3 | 392 ± 41 | 79 ± 2 | M9 | 258 ± 27 | 83 ± 2 |
| M4 | 370 ± 57 | 72 ± 9 | M10 | 457 ± 39 | 66 ± 4 |
| M5 | 173 ± 19 | 88 ± 2 | M11 | 138 ± 15 | 90 ± 3 |
| M6 | 77 ± 8 | 94 ± 2 | M12 | 80 ± 9 | 96 ± 2 |

## DISCUSSIONS

### Surface properties of fabrics

The pore size for type I masks ranges from ~80 to 460 μm. This observation suggests that the pore size in all masks is larger than bio-aerosols (≤5 μm) and in majority of masks the pores are even larger than respiratory droplets (5–100 μm). If masks having highly porous fabrics are used, then user will get reduced protection from the respiratory viruses. Recent studies have shown that the filtering efficiency of a cloth facemask has strong relation to the fabrics microstructures and suggested that fabrics having high yarn or thread packing, low porosity, and small pores can capture more particles (*Neupane et al., 2019*; *Konda et al., 2020*). In our case, among all the masks, the cotton fabrics mask M6 has smallest pore size of ~77 μm and high cover factor of ~94% (porosity ~6%). On the basis of this one can say that, out of 12 type I masks, a tightly weaved cotton facemask having low porosity (M6) could provide best filtration efficacy. The other cotton mask M12 having pore size of ~80 μm and cover factor of ~96% (porosity ~4%) could perform as good as M6. The polyester-spandex mask M10, having pore size of ~460 μm and cover factor of ~66% (porosity ~34%), will be worst of all. As in type I masks, the pores in type II masks (Table 3) are much bigger than the bio-aerosols and respiratory droplets.

It is not surprising to see different pores size and shape, yarn diameter, and cover factor even in a same fabrics type (e.g., M2, M6, M11, and M12). The surface microstructures in fabric depends on how the fabric is weaved or knitted. A tightly weaved or knitted fabric will have smaller pores and high cover factor (low porosity) than a loosely weaved or knitted fabric.

### Implications to the filtering performance of facemask

An important question one could have at this point is: *How the surface properties of fabrics can be used to screen the cloth facemask quality*? Cloth facemask are made from knitted or woven fabrics having low surface charges than electret materials used in surgical facemask and respirators (*Barrett & Rousseau, 1998*). So, the facial fit and fabric properties such as yarn packing and pore size on the fabrics are the key parameters that affect the filtering efficiency of a cloth facemask. If yarns are tightly packed then smaller pores

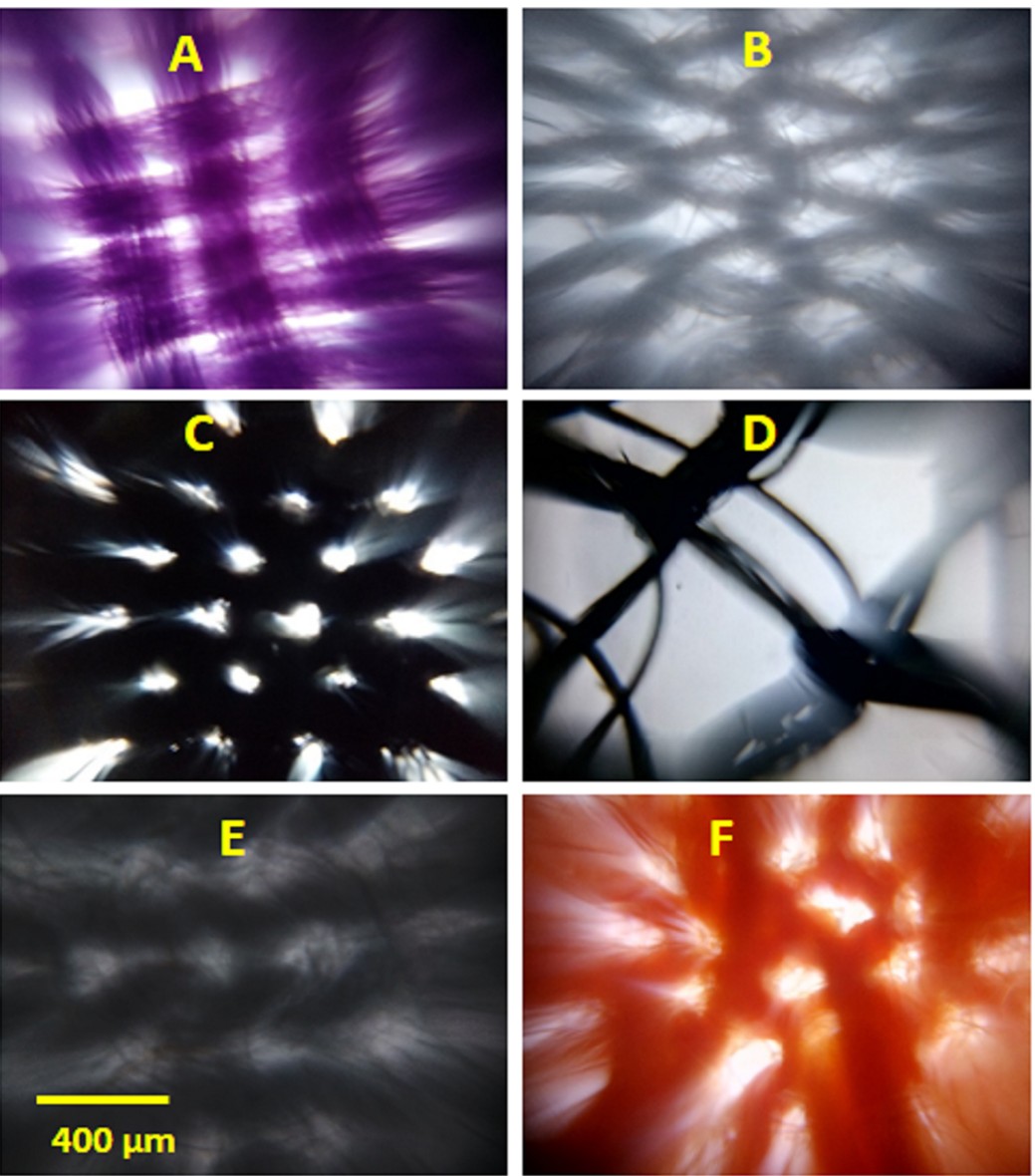

**Figure 3 Representative images of type II face masks.** (A) and (B) are the images of layer I and II for M13, (C) and (D) for M15, and (E) and (F) for M16, respectively. The scale bar shown in E is 400 μm and applies to all images.                                                            

**Table 3 Pore size and cover factor in fabrics in layer I (L1) and layer II (L2) of type II masks.** The mean pore size and cover factor was calculated form ten and five measurements, respectively.

| Masks | Layer I pore size (mean ± σ) μm | Cover factor (mean ± σ) | Layer II pore size (mean± σ) μm | Cover factor (mean ± σ) |
|---|---|---|---|---|
| M13 | 187 ± 14 | 87 ± 2 | 336 ± 21 | 76 ± 2 |
| M14 | 245 ± 15 | 71 ± 2 | 374 ± 19 | 68 ± 2 |
| M15 | 172 ± 16 | 81 ± 3 | 723 ± 20 | 42 ± 6 |
| M16 | 216 ± 15 | 83 ± 3 | 179 ± 38 | 80 ± 6 |
are formed. Recent studies have shown that the cloth facemask designed from fabrics having smaller pores and high thread or yarn packing can filter more particles (*Neupane et al., 2019*; *Konda et al., 2020*). It means, the information on the yarn or thread packing and pore size can be indirectly used for screening the cloth facemask performance. In our case, the two layered cotton fabric masks M6 and M12 mask, having relatively smaller pores and high yarn packing, is expected to perform better than other masks.

Let us say, in pandemics, someone wants to design facemask from locally available fabrics. In resource limited settings and in pandemics one may not have standard instrument to assess the quality of fabrics. In such case, the available fabric materials can be inspected for relative pore size and yarn packing using smartphone microscope demonstrated here. Although, we mounted the ball lens to the smartphone camera for image analysis, screening can also be done directly with eye in presence of room or day light. Due to small working distance of ball lens (*Cybulski, Clements & Prakash, 2014*), eye should be virtually touching the ball lens surface to see the image. After inspection, the fabric having smallest pore size and highest yarn packing can be selected for mask design. Such fabric along with improved design for better facial fitting could provide better filtration efficacy to the user.

We demonstrated an affordable fabric selection method for a better performing cloth facemask. It is known that cloth facemasks provide low breathing resistance to the user. However, it would be interesting to study the breathability test of the recommended facemask. It would also be interesting to make a systematic study on fabrics surface properties and the virus filtration efficacy of the masks.

## CONCLUSIONS

To summarize, we developed a smartphone microscopic method that can image the cloth facemask fabrics. We examined the surface of 16 different cloth facemasks and found diverse yarn packing, yarn diameter, porosity, and pores size and shape in the fabrics. Interestingly, we found that the pore size in the fabrics in the range of 80–720 μm; much larger than respiratory droplet and bio-aerosol. This partly explains why such cloth facemasks provide reduced filtration efficacy to the user. We recommend a tightly weaved or knitted fabrics having low porosity for designing a better performing cloth facemask. We believe that the method demonstrated here can be an affordable and rapid method for selection of better fabrics for cloth facemask during pandemics.

### Funding

The authors received no funding for this work.

### Competing Interests

The authors declare that they have no competing interests.

## Author Contributions

- Bhanu B. Neupane conceived and designed the experiments, analyzed the data, authored or reviewed drafts of the paper, and approved the final draft.
- Ravindra K. Chaudhary performed the experiments, analyzed the data, prepared figures and/or tables, and approved the final draft.
- Amita Sharma analyzed the data, prepared figures and/or tables, authored or reviewed drafts of the paper, and approved the final draft.

## Data Availability

The raw data are available in the Supplemental Files.

## Supplemental Information

Supplemental information for this article can be found online at http://dx.doi.org/10.7717/peerj.9647#supplemental-information.

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
