# Peer review of "A smartphone microscopic method for rapid screening of cloth facemask fabrics during pandemics"

_PeerJ, doi:10.7717/peerj.9647_

## Round 0.1 · original submission · Major Revisions

· Academic Editor

Major Revisions

Please be sure to address every comment from all three reviewers. I will be carefully looking to see if you have fully addressed each comment, and at this point, I cannot provide assurance of acceptance of your manuscript.

Reviewer 1 ·

Basic reporting

Neupane et al report on the use of smartphones to image the pore size of different fabrics. The paper is essentially saying that one can use a camera to image a fabric, which is not at all novel. Their experimental design is also not new as the authors have previously reported their setup in this publication - https://www.biorxiv.org/content/10.1101/2020.04.09.035147v2.abstract

Experimental design

The paper lacks a number of details if anyone wants to do this experiment at home - the type of smartphone used; its camera specification; how was the scale determined; fabric details, etc.

The ball lens was a commercial product and something that is not typically present in a household. During this pandemic, it may be difficult to purchase such a lens quickly. Instead, it may be easier to get your hands on a high magnification camera. So I don't agree that a ball lens + camera is more useful than a camera on its own.

Validity of the findings

The authors mention that the bright regions in the optical micrographs are pores. Still, they do not report on the pore size of M12 and M16 (first layer). Their claim that they can be used as a face mask is hence not supported by evidence. My main criticism is that they are not reporting on the kinds of fabrics they have used. They are just labeling it as M1-16. How are the general public supposed to know which fabric to use?

Reviewer 2 ·

Basic reporting

Introduction / need to add some background references (see below)
Line 83; …. include 'face seal' and use compliance. See Bard et al 2019 / https://pubmed.ncbi.nlm.nih.gov/31036302/

Line 86; From biosafety POV, the authors need to add precautionary advise for the user in handling potentially SARS-CoV-2 / pathogens contaminated mask - see link for citing WHO guidance on this file:///C:/Users/kijaz/Downloads/WHO-2019-nCov-IPC_Masks-2020.3-eng.pdf
Line 93; … virus filtering efficacy in the lab …. cite ref, suggest - Zhou et al 2018 / https://pubmed.ncbi.nlm.nih.gov/29707364/

Line 107; The authors should provide clear distinction between 'protection' and 'pathogens / particular aerosol filtration by mask materials' - so please replace 'protection' with 'filtration efficacy' throughout the manuscript.
Line 110; suggest replacing .'cheap' with 'affordable'

Experimental design

Material & methods

Line 121; provide reference for aluminum.

Validity of the findings

Conclusion

What's the pore-size (confirmed by filtration efficacy) using this 'smartphone microscopic'-screening method for mask-fabric selection the authors would recommend?

Additional comments

Given the ongoing SARS-CoV-2/COVID-19 pandemic, this “smartphone microscope method for rapid screening of fabric quality” developed by the authors provide an affordable technique to select appropriate fabric for masks particularly in resource-challenged regions. I would recommend this be considered for publication pending the minor revisions highlighting the limitations of current work and recommendations for further investigations to validate the fabric-selection based on the technique developed here (e.g., filtration of pathogens / particulates / PM2.5 needs to be carried out).

Reviewer 3 ·

Basic reporting

Lines 89-90: please correct the citation format.
Lines 94-95: is 50-70% efficiency for virus?
Line 95: what does SMS stand for?
Lines 96-97: please remove ‘\’.

Line 127-128: can you provide relevant literature on the application/design of such smartphone microscope?
raw data is not shared.

Experimental design

Were there any replicates for masks or only one masks were tested? Can authors mention more about the type of fabrics used in type I and II? Are there any differences among the masks within type I or type II? It'd be helpful to provide the pictures or more details about these masks.
Line 146: ‘Around 5-10 images …’
What could be the differences in findings with only 5 images taken vs. 10 images taken? When multiple images were taken, were they taken from the same area of the mask?

Figure 1c. Please provide more details on (c) calibration grid and how it was used for calibration.

Validity of the findings

Instead of showing only representative figures, I suggest authors to include all figures (in supplementary section).
Table 1. Why is there missing data in M6 and M12, and M16 in Table 2?
Line 219: Authors mention about high thread packing for M6 and M12? It’ll be helpful to include such information for all the masks rather than only these two types.

Additional comments

Line 105: previous paragraph suggest that cloth facemasks work fairly good (50-70%) compared to other mask types (>85%). Won’t improper facial fit also be the problem for other type of masks?
Line 106: can you please mention the values for thread density and pore size?

Line 200: based on this study, authors mentioned the nominal protection but have cited the papers showing 50-70% efficiency in cloth facemasks. Please discuss.
Line 204: I’d recommend to include pictures of all masks and mention what is different about M6 and M12 so that it’ll be more useful for readers.

---

## Round 0.2 · Minor Revisions

· Academic Editor

Minor Revisions

Your manuscript has elicited a positive response from the two reviewers who provided comments. I ask that you address the additional comments from Reviewer 3, and in your response to the reviewer please indicate changes made in the manuscript. Please be sure to address all comments.

Reviewer 2 ·

Basic reporting

Suggested literature cited

Experimental design

Details added

Validity of the findings

Conclusions are well stated

Additional comments

I think authors have addressed the comments.

Reviewer 3 ·

Basic reporting

Page 10, Lines 165-169: To avoid confusion, different terms should be used for equations (1) & (2)

Page 10, Line 174: Does ‘f’ in equation (3) refers to ‘f’ in equation (1) or (2)?

Experimental design

I suggest to move lines 178-187 to Materials and Methods.

Lines 208-209: I suggest to avoid redundancy (see lines 197-198)

I suggest to clearly mention if the authors used the same face mask for replications or different face mask (of same type) for replication.

Validity of the findings

I suggest improving results section and add major findings in this section. Several sentences in this section are likely to be more appropriate in method section.

Rather than saying, results are given in figures and tables, I suggest authors to mention their observations from these figures and tables.

In response, authors mentioned ‘For a particular mask, we found all the images very similar so we believe that our conclusion remains the same.’ Doesn’t it vary based on how tightly it is knitted and could it vary for the same cloth type?

Table 1. Authors were not able to show the pore size information for M6, M12, and M16 in the first version, but were able to show the information in the second version with contrast adjustment. I suggest to mention limitation due to this adjustment.

Additional comments

Page 4: Spelling error for author affiliations: ‘Institue’ and ‘Sceicnes’

Lines 229-230: Describing M6 mask would benefit the readers who will be looking for appropriate cloth type.

Lines 237-238: Does this mean that the results could greatly vary depending on the different samples of cloths studied under the microscope? Won’t it make difficult to make conclusion about which cloth type would be appropriate for protection?

I suggest to include the pictures of all masks in supplementary section.

---

## Round 0.3 · accepted · Accept

· Academic Editor

Accept

Thank you for your revisions, which have meaningfully improved the manuscript.